# “All You Need Is Love” a Social Network Approach to Understanding Attachment Networks in Adulthood

**DOI:** 10.3390/bs14080647

**Published:** 2024-07-26

**Authors:** Junnan Tian, Harry Freeman

**Affiliations:** 1Department of Psychology, Wuhan University, Wuhan 430072, China; tianjn@whu.edu.cn; 2Division of Counseling and Psychology in Education, University of South Dakota, Vermillion, SD 57069, USA

**Keywords:** social networks, attachment bonds, social network analysis, romantic relationships, measuring attachment

## Abstract

This study examined five dimensions of attachment network structure in a large sample of adults (*n* = 930, 57% female) between 24 to 80 years of age. We employed a newly validated diagrammatic measure, the web-based hierarchical mapping technique (WHMT), to measure the attachment strength to participants’ five closest relationships and the physical distance to and between network members. Our first aim was to replicate existing research on the composition of adult attachment networks, exploring variations in network patterns by age, romantic status, and parental status. Our second aim was to explore four new network dimensions, including physical distance to network members, hierarchical patterns, centrality, and density. The results replicated previous work on network composition, highlighting the pivotal role of romantic partners as primary attachment figures through adulthood. The analysis of the new network dimensions revealed a clear divide between adults in romantic relationships and those who are not. Compared to the single adults, the adults in romantic relationships were more hierarchical in their attachment preferences, reported lower emotional connection to friends and parents, and lived farther from their network, which was also more geographically dispersed. In other words, romantically involved adults put more of their attachment eggs in one basket. The results also showed that the older adults tended to live further away from their attachment network and had a more geographically dispersed network compared to the younger adults.

## 1. Introduction

Attachment bonds describe a limited number of relationships across the lifespan that are biologically based, durable across time and physical distance, and critical to felt security [1,2]. Infants and young children develop their first attachment bonds to parents and potentially to other adult caregivers, but beyond childhood, the composition of a person’s attachment network expands to include close peer relationships [3,4,5,6,7,8]. Beginning in the 1980s, researchers began studying the normative development of attachment networks beyond childhood, examining how friends and romantic partners become increasingly important targets for attachment support in adolescence and young adulthood [5,9,10,11,12]. More recently, this work has expanded to the study of attachment networks through the adult years [8,13,14,15,16,17]. One aim in the current study is to replicate existing research on the composition of adult attachment networks using a newly validated diagrammatic measure called the web-based hierarchical mapping technique (the WHMT) [18]. The second aim is to move beyond the current focus on network composition to study whole network dimensions, including centrality, density, and hierarchical patterns, an aim that is made possible by the new analytics of the WHMT.

Attachment network research has been limited to studying composition due to a reliance on Likert-scaled measures that use ordinal scoring, such as the WHO-TO [7], the Attachment Network Questionnaire (ANQ) [19], and the Important People Interview (IPI) [20]. These measures are well suited to assessing who in a person’s social network meets the criteria of an attachment figure and the relative strength to each network member. The ordinal scoring, however, results in non-independent ratings of attachment figures, which constrain whole network metrics that describe the shape or what we call network morphology. Network morphology dimensions include the centrality, density, and degree of hierarchical preference [18,21]. To accurately assess network centrality and hierarchy requires independent ratings of attachment strength to each network figure, and measuring network density requires independent ratings between network figures. The WHMT provides this analytic capability. In the current study, we employ the WHMT to replicate previous findings on adult attachment network composition and extend this work by studying normative patterns of network morphology from emerging adulthood to late adulthood.

### 1.1. The Study of Adult Attachment Network Composition

In discussing the ontogeny of attachment beyond infancy, Bowlby speculated that “Other adults may come to assume an importance equal to or greater than that of the parents, and sexual attraction to age-mates begins to extend the picture [1]. Theory on adult relationships as attachment bonds gradually took shape during the latter half of the 20^th^ century, beginning with Hazan and Shaver’s seminal paper (1987) [22] “Romantic love conceptualized as an attachment process”. In this paper, the authors focused on how styles of infant attachment (i.e., infancy-secure, avoidant, and anxious/ambivalent) can be applied to understanding individual differences in adult attachment to romantic partners. Later work by Hazan and others expanded this view to explain why and under what normative conditions romantic partners and other adults can take up attachment functions and become primary targets for attachment behaviors [9,10,11], culminating in a normative model of parent-to-peer attachment transfer [10]. Based on their findings, Hazan and colleagues indicate that clear-cut adult attachment formation typically occurs around the second anniversary of romantic relationships [5,10]. In addition to romantic attachment, most adults continue to rely on parents as important sources of attachment support, and often forge one or two additional attachments to friends or others. Attachment network research has focused on understanding variations in who adults turn to for attachment support and the relative strength of attachment to each network figure.

Network composition research examines who and how many people are in a person’s attachment network, and the relative strength of attachment to each network member. Determining who in a person’s social network is a legitimate attachment figure, network members are evaluated against three cognitive/behavioral attachment provisions, including safe haven, proximity maintenance/separation protest, and secure base exploration [7]. When a network member meets all three cognitive/behavioral criteria, they are said to be a “clear-cut” attachment figure [23]. The number of clear-cut attachment figures is limited, typically ranging between one and three individuals at any point in a person’s lifespan [10,21,23,24,25]. On the other hand, this determination may be partly owed to a reliance on ordinal rating scale methods that constrain the number of attachment figures that can be nominated for any single relationship provision [21]. In the current study, we revisit this question on attachment network size using a method that provides independent scoring.

The majority of attachment network research has focused on who adolescents and adults prefer as their primary source of attachment support, called the primary attachment figure. Beyond childhood, three factors have emerged as the most potent indicators of who adolescents and adults view as their primary attachment figure. These include romantic relationship status [5,10,26,27,28,29], parental status [16,30,31], and the transition into late adulthood [15,16]. From early to late adolescence, youth form more intimate friendships and enter into their first romantic relationships, a period recognized as the parent-to-peer transfer in attachment preference [10,26,27]. During this time, romantic partners and friends gradually take up attachment provisions from parents; however, a primary preference for a peer is typically delayed until two years into a romantic relationship [5], albeit a rapid transfer to a peer, friend, or romantic partner, is more likely for youth with insecure parental attachments [6]. From early adulthood and beyond, however, being in a committed romantic relationship is typically associated with lower attachment to mothers, fathers, and friends, independent of romantic relationship duration [14,16,32].

The transition to parenthood marks another developmental shift in network composition. In separate studies yielding somewhat contradictory findings, Feeney and colleagues investigated how becoming a parent is associated with differences in the attachment preference to parents and romantic partners [16,31]. In her first study, new parents indicated a stronger preference for parents and lower romantic attachment compared to their childless counterparts [31]. In a second and larger study with parents of older children, being a parent was associated with weaker attachments to parents. The authors interpreted conflicting findings as a matter of new versus experienced parents and parenting infants versus older children, respectively [16]. That is, most experienced parents of older children have come to rely on their spouses, relinquishing parents as primary confidants.

The transition to late adulthood brings additional changes due to network attrition, resulting from parents’ death or incapacity, romantic dissolution, or the death of other family members and friends. Adult children become increasing targets for attachment support, and friends and siblings are commonly identified as primary attachment figures among single adults [15,16,17]. In the current study, we explore these same factors as predictors of network composition through the adult years, using a new measure. 

### 1.2. The Measure of Attachment Networks

Traditional approaches to measuring attachment networks have relied on six- to nine-item ranking scales that capture the relative strength of attachment to significant others, based on safe haven, separation protest/proximity maintenance, and secure base exploration behaviors and cognitions [7]. The first published attachment network measure, the WHO-TO [7], was initially developed as a single forced-choice scale and was later modified by Fraley and Davis [5] for Likert scaling. Similar measures followed, including the Attachment Networks Questionnaire (ANQ) [19], and the Important People Interview (IPI) [20], which used six- and nine-item ranking scales, respectively, to assess the same three dimensions of attachment support.

All three scales—the WHO-TO, ANQ, and IPI—provide ordinal scoring of attachment strength to each rated figure, identifying who in the network meets the criteria for “clear-cut” attachment and who is the primary attachment figure. For a social network member to be considered a clear-cut or “full-blown” attachment figure, they must be ranked first or second on at least one item on each of the three dimensions [16], with the primary attachment figure receiving the highest sum-score. Network figures that do not meet this cut-off score may still be considered a subsidiary attachment figure if they are identified on one or more attachment provisions. The ordinal scoring, however, constrains accurate within-subject comparisons between network figures, given the lack of independence between ratings [18,21].

Moving beyond the measure of network composition. An alternative approach to ordinal scoring came in 2005 with the Bull’s Eye diagrammatic measure [33], a modification of Antonucci’s [34] hierarchical mapping technique requiring participants to place self-selected support figures on a diagram relative to a “core-self”, marked in the center, with three concentric circles. This new approach provides independent and continuous rating scores for each network figure, thereby providing more analytic capability for accurate comparisons between network figures [18]. The Bull’s Eye also came with some limitations, including manual delivery and scoring and low discriminate validity [18]. It was originally designed for paper delivery, using self-adhesive dots for support figures and requiring a ruler to score distances. More importantly, when participants evaluate their social network for closeness to “core-self”, intimacy and companionship are sometimes evaluated more strongly than attachment support functions, resulting in elevated peer relationship scores [18].

In response to these limitations, the Bull’s Eye was modified for online delivery and the middle “core-self” was replaced with “vulnerable-self”. Participants were instructed to think about how close they want each network figure when they are feeling insecure and vulnerable. These significant changes resulted in a new measure, which we called the web-based hierarchical mapping technique (WHMT) [18,21]. A psychometric examination of the WHMT captures has shown accurate and reliable capture of safe haven, separation protest, and secure base provisions in adult relationships with parents and romantic partners. The measure is also sensitive to micro-level changes in felt security following interventions [18]. Similar to the Bull’s Eye, the WHMT provides independent ratio-scaled scores of emotional closeness and physical proximity to each network figure, as well as between network figures. Compared to traditional ordinal formats, the independent scoring provides a more accurate assessment of each figure’s role as an attachment figure. In a qualitative evaluation of the WHMT, the authors found that support figures placed inside the first concentric circle met the definition of clear-cut attachments, indicated by separation protest, safe haven, and secure base themes [18].

In addition to a more accurate assessment of network composition, the WHMT provides whole network analytics, including network density, centrality, and hierarchical shape. The concept of attachment hierarchies is central to Bowlby’s assumption that all infants and children are innately predisposed to orient toward a preferred or primary attachment figure, a concept he termed monotropy [1]. Monotropy is difficult to test using ordinal scales, given the dependency of scores between figures. Studies using ordinal scales assume hierarchical order by assigning a primary attachment figure status to the figure with the highest sum score [5,16,19]. One exception is a recent study by Freeman and Simon [21], who applied a significance test to the difference score between the three highest rated figures. They found that 10% of their young adult sample did not significantly differentiate between their top three rated figures, and an additional 25% viewed their top two figures equally, a condition Schaffer and Emerson [35] referred to as “shared primary attachment” in their 18-month-old sample. In the current study, we examine if a clear order of attachment preference exists (i.e., monotropy), compared to having two primary attachment figures (i.e., shared primary), or if all three are undifferentiated, a condition we call distributed attachment based on Freeman and Simon’s [21] classification scheme.

In addition to testing hierarchical shape, we explore two additional social network dimensions, including centrality and density. Centrality and density are the most common outcomes assessed in social network analysis (SNA); however, they have yet to be applied to the study of adult attachment networks [36]. In SNA vernacular, attachment networks can be regarded as egocentric networks, also called personal networks, as they are comprised of an ego (the attached), alters (attachment relationships connected to the ego), ties between the ego and alters, and ties between alters [37]. Centrality is the strength of the ties connecting the ego to all the alters, and density is the strength of the ties between alters. Centrality is evaluated in terms of attachment strength to each member of the network based on how close each figure is placed to the ego. Given that participants are not asked to place alters in relation to other alters, only in relation to themselves, we do not calculate network density based on emotional closeness between network members. However, network density and other network metrics are captured in a second WHMT diagram, discussed next.

In the current study, we have participants complete a second WHMT diagram, in which they place the physical location of each attachment network figure. In this diagram, the center represents the participant’s home, or place of residence, and each of the three concentric circles represents a specific distance from the center: 10 miles, 150 miles, and 500 miles, respectively. The diagram is oriented with north, south, east, and west markings. Participants are instructed to place each network figure into the diagram to show the distance and direction to each figure and the distance and direction between network figures. The result is a two-dimensional map of a person’s attachment network with physical distance scores to each network figure and between every pair of network figures. The between network figure scores enables the researcher to calculate the physical density of a person’s attachment network, in addition to centrality.

### 1.3. Current Study

Existing studies on adult attachment networks are limited by a reliance on ordinal measures and consequently provide a limited view of network structure that focuses on network composition and on the primary attachment relationship. The current study greatly expands this approach by using the ratio-scored WHMT to measure five dimensions of adult attachment network structure, including composition and physical distance, as well as three dimensions of network morphology, including hierarchy, centrality, and density. The first aim is to replicate existing findings on attachment network composition. Three hypotheses are proposed: 

**Hypothesis** **1.** 
*Most adults will have, on average, two clear-cut attachment figures, with a minority of adults reporting no clear-cut attachments.*


**Hypothesis** **2.** 
*Romantic partners will be the primary attachment figure through the adult years.*


**Hypothesis** **3.** 
*Age, parental status, and/or romantic status will be related to the strength of attachment to friends and parents.*


The second and principal aim of the current study is to extend existing research by exploring new dimensions of adult attachment networks. Four new network dimensions are explored, including physical distance, hierarchy, centrality, and density. Given the lack of previous work in this area, we do not propose specific hypotheses. Instead, we explore these three dimensions as research questions, namely whether hierarchy, centrality, and density are associated with age, parental status, and romantic status.

## 2. Method

### 2.1. Sample

Participants included 930 adults (57% female) between 24 and 80 years of age (*M* = 42.9) living in the United States. The majority of participants (77.5%) identified as Caucasian, followed by African American (9.4%), Asian (6.9%), Latino (6%), and the remaining 0.2% identified as other ethnic groups. Based on income and education background, 14.8% of the sample was classified as working class, 48.7% as lower-middle class, 25.8% as middle class, and 10.7% as upper-middle class. 

Given our focus on developmental trends in network structure through the adult years, we examined if sample demographics (i.e., gender, romantic status, and parental status) varied significantly by age (see Table 1). Results indicated a similar distribution of gender and romantic status within each age groups. Although the number of people who were married or engaged increased slightly from early adulthood to early middle age, the difference was not statistically significant. Not surprisingly, early adults (22–34 years old) were significantly less likely to be parents compared to older adults, *t*(923) = 5.34, *p* < 0.001. 

### 2.2. Procedure

Participants were recruited through Amazon’s Mechanical Turk (MTurk), in conjunction with Cloud Research, a third-party website that interfaces with MTurk through their proprietary toolkit. MTurk is an online crowdsourcing platform where researchers can recruit a diverse and large pool of participants, called MTurk workers, to complete tasks, including surveys and experiments. We used the Cloud Research toolkit to screen MTurk workers on inclusion criteria. Our target population consisted of adults with some residential stability in their lives so that the physical closeness to their social network would vary widely as a function of a recent transition or living in university housing. As such, we stipulated that participants be at least 24 years of age and have lived in the same residence in the USA for the past year. In addition, we used Cloud Research to vet the quality of Mturk participants based on several indicators linked to valid and reliable crowdsourcing data, including HIT acceptance rate (the tasks that require human intelligence and input that are accepted by participants or workers), reliable completion of attention checks, and past completed HITs. A HIT (Human Intelligence Task) refers an Mturk worker who enrolls in the study and completes the study task. Employing these Cloud Research quality control tools has been shown to significantly improve data quality compared to those available on Mturk alone [38]. Once screened and having given consent, participants completed two online measures, including a 10-min WHMT diagrammatic measure, followed by a 10-min Qualtrics survey. Participants who completed both measures and passed attention checks were compensated USD 2.25.

### 2.3. Measures

Network structure. To measure the attachment network structure, participants completed two versions of the web-based hierarchical mapping technique [18]. The WHMT is a diagrammatic online measure that begins by asking participants to self-select the five most important relationships in their current life from a dropdown menu of 17 possible relationships (e.g., mother, father, sister, aunt, best friend, etc.), including a write-in option. After selecting, participants are taken to a second screen, displaying a target diagram with “you” marked in the center and surrounded by three concentric circles (see Figure 1). The five selected relationships now populate a column outside the diagram, with each relationship represented as a circle icon. Participants are instructed to drag and drop each icon into the diagram so that the distance to the center represents “how close you want to be to that person when you are feeling emotionally insecure, unprotected, or vulnerable.” Once submitted, pixel distance between the center and each person is recorded, with scores ranging from 0 (center) to 700 (outside the third circle). Lower scores indicate stronger attachment.

Next, participants completed a second version of the WHMT based on physical distance to and between the same five relationships (see Figure 2). In this second diagram, the three concentric circles represent 10 miles, 250 miles, and 500 miles from the center, marked “home”. The diagram is marked with north, south, east and west so that participants can place where each network figure lives geographically in relation to the participant. Participants are instructed to drag and drop each relationship icon into the diagram so that it represents where each person lives relative to the participant and relative to the other network figures. Upon submission of the completed diagram, pixel distances are recorded to each network figure from the center and between each network figure. Pixels distances were transformed into miles to represent physical distances to the participant and between network figures. 

Based on the two WHMT diagrams, we calculated the network composition and three dimensions of the network morphology, including network hierarchy, network centrality, and network density. The measure of each network variable is detailed next. 

Network composition. Network composition was measured by identifying who participants selected as their five closest relationships and their attachment strength to each network figure, calculated as the pixel distance to the center of the diagram. Shorter distance corresponds to stronger attachment. In a recent study examining the validity of the WHMT as a measure of attachment strength [18], it was found that participants who placed network figures in the inner circle (60 pixels from the center) consistently met the criteria as clear-cut attachment figures. Using 60 pixels as the cut-off criteria, the number of attachment figures was calculated.

Network morphology (shape of the network). Three dimensions of network morphology were determined, including the degree of hierarchy, network centrality, and network density. Network hierarchy describes the extent to which participants perceive a clear order of attachment preference to a primary target for attachment support. We examined this by comparing the difference in attachment strength between the primary and secondary figure and between the primary and tertiary figure, resulting in three ordinal classifications. The first category, monotropic, was marked by the presence of a clear primary attachment figure, who was rated significantly stronger than all the other network figures. A second category, shared primary, was marked by the top two network figures being undifferentiated, but both figures were rated significantly stronger than the remaining network figures. The third category, distributed, was marked by the top three figures, having relatively equal ratings of attachment strength [21]. The network structures were converted and determined based on a cut-off value that indicated a meaningful difference between attachment figures. A clear order of preference between figures was classified based on a clear visual difference in relative position to the center, which was based on the validation study [18]. Each figure icon equaled 18 pixels in radius, and this distance was used as the cut-off score for a visually clear distinction in closeness to center.

The second morphology dimension calculated was network centrality, which corresponds to the average distance of each network figure to the center. Centrality was calculated as the average pixel distance of the five selected close relationships. We chose to include all five figures, given that non-clear-cut attachment figures are often relied on as subsidiary attachment figures under certain conditions.

The third morphology category, network density, refers to the closeness of network figures to each other. This metric was not calculated for attachment strength, given that participants were not asked to place figures in the first WHMT diagram relative to how close network figures were to each other. For this metric, we used the second WHMT diagram that provided information on the physical location of each network figure relative to the participant’s home, and relative to the other network figures. Pixel distances were converted to miles, the average distance to the center was computed as the physical centrality of the network, and the average distance between network figures was computed as physical density.

Demographics and normative life events. Following completion of the two WHMT diagrams, participants completed a Qualtrics survey that included the following demographic and life event questions. The descriptions of romantic status included six options: “Not currently romantically involved with someone or dating”, “Dating one person but we are not seriously involved”, “Dating one person and we are seriously involved, but we are not living together”, “Living with my partner, but we are not married”, “Engaged to be married”, and “Married”. For the analysis, options were collapsed into three categories, including single, dating, and married/engaged. For the parental status, participants were asked, “Do you have children, if so, how many children do you have?”, and the parental status was eventually coded as a dichotomous variable (0 = no child and 1 = child).

To assess the socioeconomic status (SES), participants were asked about their education background and annual income. We followed Wani’s [39] recommendations for weighting these indicators. The total score of income and educational level was calculated, and the average score of the two levels was used and converted into a combined SES.

## 3. Results

### 3.1. Descriptive Statistics

Bivariate associations were run between network structure variables and demographic characteristics (see Table 2). Age was significantly and positively associated with parental status, attachment strength to mothers, physical centrality, and physical density, indicating that older participants were more likely to have children, report lower attachment strength to mothers, and to have a social network that, on average, was further from their home (centrality) and more spread apart (i.e., density). Gender was significantly and positively correlated with parental and romantic status, indicating that females were more likely to report having children and being in a romantic relationship. In addition, attachment strength to mothers and romantic partners were both significantly and positively connected with physical centrality and density, as well as attachment strength centrality. Participants with stronger attachments to mothers or romantic partners had a network of figures that tended to live closer, tended to live closer to each other, and tended to provide greater combined attachment support. Interestingly, romantic strength was negatively associated with attachment strength to parents and friends, whereas attachment to fathers, mothers, and friends were all positively related.

### 3.2. Network Composition

In the first set of inferential analyses, we focused on replicating the existing work on adult network composition. The hypotheses and analytic strategy are primarily based on the work by Doherty and Feeney [16], who examined an adult sample of similar size and similar age range.

Hypothesis 1: most adults will have, on average, two clear-cut attachment figures, with a minority of adults reporting no clear-cut attachments. Network figures were classified as a “clear-cut attachment” if placed inside the inner circle of the WHMT diagram, a method that was validated based on extensive interviews in a recently published study [18] (see also methods/measures/network composition for more information on how we made this determination). 

The results indicated that having two clear-cut attachment figures was average (M = 1.96, SD = 1.2) and the most frequent outcome (33.3%). Nearly a third of participants identified a single clear-cut attachment figure (30.3%), followed by participants reporting three clear-cut figures (18.2%). In sum, 81.8% of participants had one to three clear-cut attachment figures. As predicted, a minority of participants, 8.2% of the sample, had no clear-cut attachment figures, a classification based on placing all five self-selected support figures outside the inner circle. Another 10% of the sample identified either four (5.2%) or five (4.8%) clear-cut attachment figures. 

Next, we examined developmental trends associated with attachment network size. The bivariate correlation between age and number of clear-cut attachment figures was close to 0 (*r* = 0.012). We ran a one-way ANOVA to examine if differences existed between the four age groups (24–34; 35–44; 45–64; ≥65), which also indicated no developmental trends in network size (F(3, 927) = 0.82, *p* = 0.466).

Hypothesis 2: romantic partners will be the primary attachment figure through the adult years. The primary attachment figure is typically identified as the highest ranked figure. However, this approach applies to studies using traditional Likert-scaled measures, in which independent comparisons between network figures is hampered by ordinal scaling. In the current study, participants indicated their closeness to each figure separately, which allowed for more than one primary figure to be indicated. We classified the first primary figure as the one with the closest pixel score to the center. Given that most individuals have less than three attachment figures, we allowed for up to three primary figures. A second or third primary figure was added if these figures were placed within 15 pixels from the top-rated figure. The 15-pixel cut-off score corresponds to a distance that is visually difficult to differentiate (see Method and the measurement of network hierarchy for more details on this method). Table 3 shows the percentage of each relationship type nominated as a primary attachment figure within each age group. Given that individuals can select more than one primary attachment figure, when all the figures are summed, the total exceeds 100%. 

The results in Table 3 clearly show the primacy of romantic partners as primary targets for attachment support through the adult years. Romantic partners are nominated twice as often as the next most identified figure, mothers, and three to four times more often compared to the third most rated figures (siblings, friends, and fathers). A series of one-way analysis of variance (ANOVA) procedures were run for each relationship type to examine if the likelihood of being nominated as a primary figure varied between the four age groups. The results were significant for mother (F(3, 926) = 3.86, *p* = 0.009), father (F(3, 926) = 2.82, *p* = 0.038), and child (F(3, 926) = 11.25, *p* < 0.001). Older adults (late middle age and late adulthood) were less likely to nominate mothers and fathers as primary figures compared to early adults or early middle-aged adults. However, the nomination of children as primary figures steadily increased from early adulthood to late adulthood.

Hypothesis 3: age, parental status, and/or romantic status will be related to the strength of attachment to friends and parents. To examine if age, romantic status, and parental status are associated with attachment strength to different relationship types, a series of sequential regression analyses were run. Age, parental status, and romantic status were entered as factors in the first block, and all possible two-way interactions among these variables were added in the second block using centered scores.

The first model of main effects was significant for all relationship types except siblings (see Table 4). Romantic status was statistically and positively predictive of strength to mothers, fathers, and friends, indicated by a stronger attachment to parents and friends among the single participants. Age was statistically and positively predictive of strength to mothers and children. The result suggested that younger adults had stronger attachment strength to their mothers compared to older adults. Surprisingly, the results indicate that the attachment strength to children diminished slightly with age. This finding is counterintuitive and contrasts sharply with Doherty and Feeney’s [16] findings. This result may be an artifact of how child strength was included in the networks. “Child” was not a category in the list of available relationships, thus requiring participants to write in this category, which was done so by a minority of participants with children. If we examine who wrote in a child as an attachment figure by age, the significant bivariate correlation (0.225) indicates that older adults were significantly more likely to list a child as a network member. In addition, as presented under hypothesis two, age was positively related to whether the participant viewed their child as a primary attachment figure. Consequently, these findings are consistent with previous work [4,5,15,16,17]. Lastly, parental status was also a statistically significant predictor of attachment to romantic partners, indicating that the adults with children had stronger attachment to romantic partners compared to their non-parent counterparts.

### 3.3. Moving beyond Network Composition

In this study, we measured and analyzed several new network dimensions to describe more fully the structure of attachment networks through the adult years, including physical proximity to and between network figures, and three dimensions of network morphology, including hierarchy, centrality, and density. 

Physical proximity. This network dimension is based on the second WHMT diagram, in which participants placed their five self-selected important relationships in a geographical location in relation to their own home, and thereby, in relation to the other network figures. The average pixel distance of physical proximity to self and to other network figures was transformed and converted to miles (see Table 5). The participants had the closest physical distance to romantic partners, which is not surprising, given that 73% of those in romantic relationships were cohabitating. Of those in committed relationships but not living with their romantic partner (*n* = 172), 70.3% still lived within 10 miles of their partner.

Research question 1: what is the relationship between age, romantic status, parental status, and physical proximity to each network figure? To investigate whether age, romantic status, and parental status are related to physical proximity to each network figure, we conducted a series of sequential regressions, just as with the attachment strength analysis. The first model of main effects was all significant (see Table 6). 

Age was statistically and positively predictive of physical proximity to romantic partners, mothers, and children. The results suggest that older participants, on average, lived further from these figures. Furthermore, parental status statistically and negatively predicted physical proximity to romantic partners, which suggested that having children was related to close physical distance to romantic partners. Finally, romantic status was statistically and positively predictive of physical proximity to mothers, fathers, siblings, and friends. Those who did not have romantic partners had close physical distance to parents, siblings, and friends.

Morphology of attachment network. We examined three dimensions of network morphology, including hierarchical structure, centrality (for both attachment strength and physical distance), and density (for physical distance only). Hierarchical structure was classified into three variations, including the monotropic, shared primary, and distributed attachment categories [21]. Monotropic structure is based on Bowlby’s [1,2,3] assumption of “monotropy”, which suggests that humans are predisposed to have a clear primary attachment figure marked by meaningful separation in attachment strength between the highest rated figure and all the other network figures (see the measurement of hierarchy under Method/Measures). Shared primary structure is marked by an undifferentiated rating of the top two figures, which are both rated significantly stronger than the remaining three network figures. This term was used by Schaffer and Emerson [35] to describe the undifferentiated attachment to mothers and fathers that many of their infants showed by 18 months of age. The third classification, distributed attachment, was used by Freeman and Simon [21] to describe a lack of preference between the top three rated figures. The results demonstrated that 69.8% of the sample had the monotropic structure, followed by the shared primary structure (17.2%), and distributed attachment (13%).

Research Question 2: is age, romantic status, or parental status related to the degree of network hierarchy between support figures? To test effects of age, romantic status, and parental status, the same sequential regression analyses as attachment strength and physical proximity were run to predict the degree of network hierarchy between support figures, the emotional or physical centrality of network figures, and the emotional or physical density of network figures (see Table 7). Parental status statistically and negatively predicted the degree of network hierarchy between support figures. Those who have children had non-hierarchical attachment network structures, and they considered that all their support figures were equally significant. Nevertheless, romantic status was statistically and positively predictive of the degree of network hierarchy between attachment figures, which demonstrated that those who have romantic partners had hierarchical network structures. Romantic partners would more likely be identified as the primary figure.

Emotional and physical centrality and density of network figures. In addition, attachment centrality was calculated as the average attachment strength to all five figures, and physical centrality was calculated as the average physical distance to all five network figures. Physical density was calculated as the average physical distance between every possible pairing of the five network figures. Given that participants were not asked to consider the attachment strength between figures, attachment density was not calculated.

Research Question 3: Is age, romantic status, or parental status related to attachment centrality and physical centrality of network members? To test effects of age, romantic status, and parental status on network centrality (both attachment and physical) two sequential regressions were run. Results are shown in Table 7. Age positively predicted the physical centrality of network member, younger adults, on average, lived closer to their network. Parental status negatively predicted attachment centrality of network figures, indicating that participants with children had stronger attachment networks, and lived closer to their attachment network

Research question 4: is age, romantic status, or parental status related to the emotional or physical density of network figures? An additional sequential regression was run to test the effect of age, romantic status, and parental status on physical density (see Table 7). Only age was statistically and positively related to the physical density variable, indicating that older participants tended to live further away from their attachment network.

## 4. Discussion

### 4.1. Network Composition

The network composition findings based on the WHMT diagrammatic measure are highly consistent with previous work using traditional Likert scaled measures, such as the WHO-TO, the IPI, and the ANQ [4,5,15,16,17]. Attachment network size did not significantly vary by age. Most participants reported one or two clear-cut attachment figures, and about one in five adults indicated three attachment figures. Having more than three or having no attachment figures were both uncommon conditions, each describing approximately 10% of the sample. These results support the view that attachment networks are comprised of a few relationships at any one time from cradle to grave, and that entrance into the inner circle of attachment relationships is highly exclusive, supporting Ainsworth’s contention that attachments apply to a limited number of relationships [23]. Within this inner circle of attachment relationships, most adults further differentiated a hierarchy of preference. 

The adults in romantic relationships, independent of age, gender, or romantic relationship duration, overwhelmingly identified romantic partners as their primary target for attachment support. Among the single adults less than 44 years of age, mothers were the most common primary target. Mothers remained critical attachment targets into middle adulthood among many single adults between 44 and 64 years of age, being selected as often as friends as the primary figure. The primacy of romantic attachment and the longevity of maternal attachment are consistent with theoretical views on attachment across the life course [11,17,23].

These results strongly support the view that most people continue to rely on mothers as attachment figures through their adult life, and significant disruptions in maternal status are less likely the result of gradual emotional distancing than a function of romantic status, maternal loss through death, or a diminished capacity to fulfill this role. The primacy of romantic partners and mothers suggests that the process of parent-to-peer attachment transfer continues to operate in a bidirectional fashion through most of adulthood, such that a new romantic relationship quickly gives way to prioritizing romantic partners over mothers, after which mothers remain important secondary targets. Given that romantic duration did not mediate the strength of romantic attachment, the process of prioritizing partners over mothers is fairly rapid and not the gradual transfer of attachment provisions observed in adolescence or young adulthood [5,10]. Similarly, in the event of romantic partner loss or dissolution, mothers may immediately resume their original place as the primary figure. Of course, these conclusions are tentative and based on cross-sectional data that preclude a clear understanding of the process and temporal pace of romantic attachment formation. 

The shared experience of parenting appears to strengthen reliance on partners for emotional support and possibly promote deeper attachments. Feeney and colleagues [16] reported similar findings in their older adult sample but found the opposite effect for younger parents, in which case having a child strengthened their parental ties, not their romantic relationship. In the current study, the association is weak and may point to the vicissitudes in how parenting influences romantic bonds. In this way, becoming a parent may not be consistently tied to the strengthening or weakening of a specific relationship, but rather signals a redistribution of attachment that is idiosyncratic with respect to individual and contextual factors. This idea is reinforced in the next set of findings on normative life events, including child status, on network centrality.

Being romantically involved and having higher romantic attachment strength predicted less attachment strength to mothers, fathers, friends, and adult children. Beyond middle adulthood, however, other family members, including siblings and adult children, became increasingly common attachment targets. Being a parent was associated with stronger reliance on romantic partners for attachment support, but this life event was not related to attachment strength to other network figures. The adults that did not indicate a preference for romantic partners or mothers identified friends, siblings, or adult children as the most sought primary attachment figures. Fathers, on the other hand, were consistently uncommon primary targets regardless of age, with approximately one out of every twenty participants viewing their fathers in this role.

### 4.2. Beyond Network Composition

Up until now, research on adult attachment networks has focused on questions of network composition, namely, who the primary attachment figure is and how many social network figures function as “clear-cut” attachment relationships. This work has provided important insights into in size of attachment networks and who function as primary attachment figures through normative adult transitions, but composition alone provides an incomplete understanding of network structure. A major strength of the current study lies in the exploration of novel network structures using the web-based hierarchical mapping technique (WHMT). In addition to composition, The WHMT provides data on whole network dimensions, which we call network morphology, including hierarchical structure, centrality, and density. We explored composition and morphology using both attachment strength and the physical location of network members to provide a fuller understanding of attachment networks through the adult years as a function of parental status and romantic status.

### 4.3. Physical Proximity to Different Figures 

Growing older was associated with greater physical distance to and between attachment network members. This distancing is likely the result of greater mobility in early to midlife transitions, such as leaving school, entering a career, changing vocations, as well as housing transitions and relocations that accompany normative increases in family size (i.e., becoming a parent) [40]. Conversely, relocation in later adulthood may be triggered by life events that reduce household composition, such as romantic relationship dissolution or death, adult children leaving home, and downsizing one’s living space after retirement [14,15].

Greater physical centrality and network density were associated with stronger attachments to romantic partners, mothers, fathers, friends, and siblings. These findings underscore the importance of physical availability and access to our closest relationships, even in an age where technology provides instant access to our social network. When coupled with age trends in centrality and density, the developmental trend is toward increasing disconnection from our closest social network, both physically and emotionally. While it is not possible to tease apart the direction of effects, it seems more likely that physical distance due to mobility is the instigating factor for social disconnection.

Romantic status was an important indicator of physical proximity to network members. Single participants lived closer to their parents, siblings, and friends, compared to their coupled counterparts. As discussed, singlehood also corresponded to greater attachment strength to these same figures. Collectively, these findings are consistent with a common perception that spending time together in the same physical space matters more than the same amount of social interaction experienced through distance communication. On the other hand, our data do not provide this fine-grained analysis. An important next step is to document associations between physical centrality and density in relation to interaction context (distance versus in-person). Such a study could lead to a better understanding of how physical closeness directly or indirectly supports emotional closeness.

The romantic and parental status were important, albeit opposite indicators of whether adults indicated a clear preference hierarchy or not. Being in a romantic relationship was associated with a clear monotropic orientation toward one network figure, namely the romantic partner. However, being a parent was associated with a more distributed network. Put another way, romantic relationships tend to pull attachment strength from other network figures, whereas having a child appears to dampen this effect and result in a more equal distribution of attachment preference. This romantic-status-by-parental-status interaction effect was the only interaction effect in all the sequential regressions run with these two life-event factors. At the same time, this effect runs contrary to the child status effect on attachment strength, which suggests that having a child increases romantic partner strength, albeit the association is very small.

According to the attachment and physical centrality of network figures, the findings indicate that individuals tend to have stronger and more consolidated attachments to all network figures when they have children. Parenthood is a shared experience that often brings parents into contact with others who are also going through a similar life event [41]. This shared experience can foster stronger bonds between parents and their peers, as they have a mutual understanding and support system. Moreover, having children often means increased demands on time, energy, and resources. Parents often rely on their social network for practical support, such as help with childcare, advice on parenting issues, and emotional support during challenging times [42]. This reliance may deepen their connections with these individuals.

### 4.4. Limitations and Future Directions

The limitations of this study are primarily due to a restricted sample (MTurk workers) and the use of cross-sectional data. The respondents in the present study were recruited from a convenience sample of MTurk workers. Although the sample size was fairly large and distributed across the USA, the ethnic diversity was limited and non-representative of the US population, limiting the generalizability to the broader population [43] and especially to non-Western samples. A study in Bangladesh, for instance, found that parents, not romantic partners, remained primary attachment figures for adult women into middle age, an outcome the authors attributed to collectivist cultural values [44]. The reliance on cross-sectional data allows us only to speculate on questions of stability, dynamics, and change in attachment network structures across adulthood. 

An important next step is to examine how these same network dimensions, including composition and morphology, operate in diverse cultures, and how these network dimensions predict adult adjustment and physical and mental health. Ideally, such a study could follow participant networks over time to shed light on stability and change in attachment network structures and how these temporal factors are related to well-being and health outcomes. 

### 4.5. Conclusions

These results are the first to examine attachment networks using social network factors including density and centrality. The addition of geographic location of network members also adds to our understanding of how physical distance and emotional connection interrelate within the context of normative adult transitions, including romantic relationship status and parent status. Looking across the three dimensions of morphology, there appears to be a strong dividing line between adults in romantic partnerships and those who are not. Those in romantic relationships are more likely to put their attachment eggs in one basket, live further from their other close relationships, and have a network that is more geographically dispersed. In sum, the more isolated and hierarchical network among romantically involved adults fits the Beatles’[45] refrain, “All You Need is Love”. Single adults are more likely to spread their attachment preference equally between two or three closest relationships, often including a friend and a family member. Single adults also tend to live closer to their attachment network. Yet, having a child may lead to some network reorganization by forming new social connections closer to home. In addition to forming romantic partnerships, growing older also contributes to diminished network centrality and density over time. Taken together, these findings reveal a more comprehensive and nuanced look at attachment networks than previous literature has uncovered. Longitudinal research and studies with culturally diverse samples are important avenues for future research.

## Figures and Tables

**Figure 1 behavsci-14-00647-f001:**
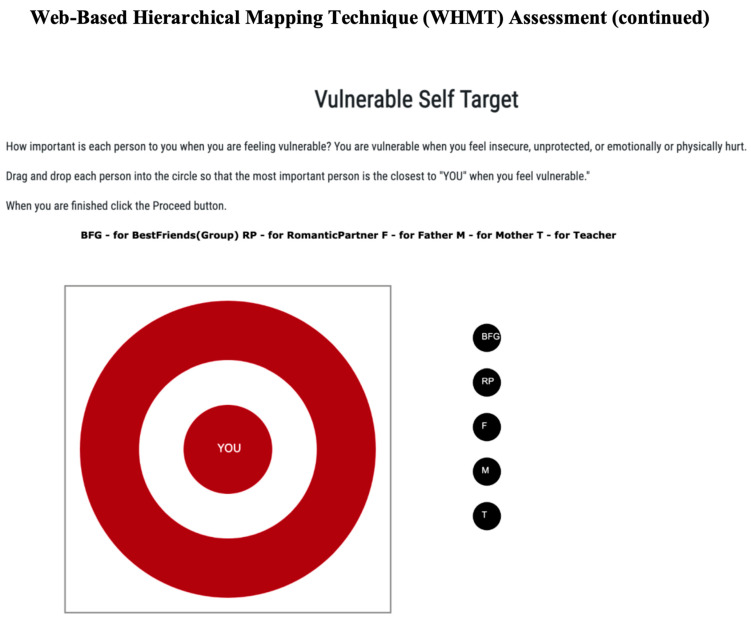
WHMT instructions—emotional distance.

**Figure 2 behavsci-14-00647-f002:**
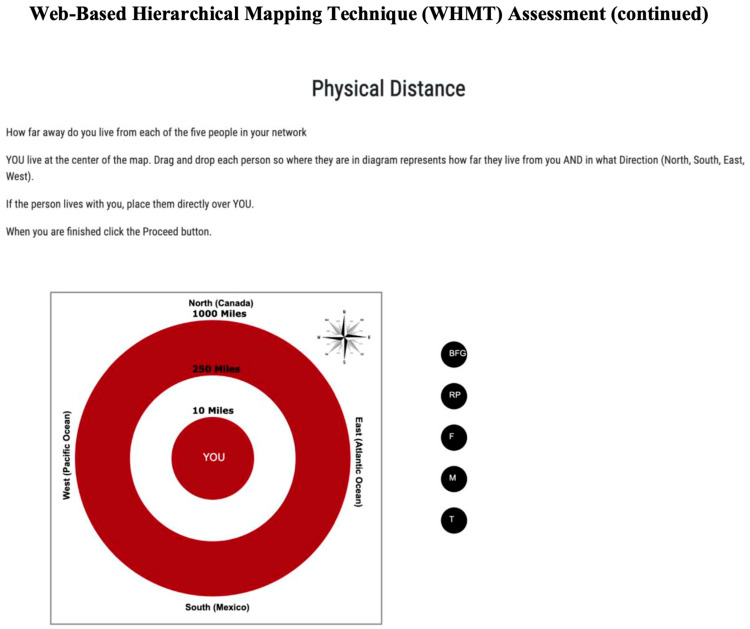
WHMT instructions—physical distance.

**Table 1 behavsci-14-00647-t001:** Demographic characteristics based on age groups.

		Gender				Romantic Status						Parental Status			
		Male		Female		Single		Dating		Married/ Engaged		No Child		Child	
Age Group	*n*	*n*	%	*n*	%	*n*	%	*n*	%	*n*	%	*n*	%	*n*	%
Early Adulthood (ages 22–34)	288	126	43.8	162	56.3	90	31.3	42	14.6	156	54.2	211	73.3	77	26.7
Early Middle Age (ages 35–44)	304	145	47.7	159	52.3	70	23.0	31	10.2	203	66.8	170	55.9	134	44.1
Late Middle Age (ages 45–64)	276	105	38.0	171	62.0	72	26.1	33	12.0	171	62.0	150	54.3	126	45.7
Late Adulthood (ages 65 and older)	62	25	40.3	37	59.7	21	33.9	5	8.1	36	58.1	36	58.1	26	41.9
Total	930	401	43.1	529	56.9	253	27.2	111	11.9	566	60.9	567	61.0	363	39.0

**Table 2 behavsci-14-00647-t002:** Correlations between demographics and network structure variables.

Variable	1	2	3	4	5	6	7	8	9	10	11	12
1. Age	-											
2. Gender	0.057	-										
3. Romantic Status	0.003	0.073 *	-									
4. Parental Status	0.123 **	0.069 *	0.489 **	-								
5. Attachment Strength to Romantic Partner	−0.007	−0.035	NA	−0.057	-							
6. Attachment Strength to Mother	0.088 *	0.056	0.101 **	0.036	−0.306 **	-						
7. Attachment Strength to Father	0.041	0.105 *	0.022	−0.034	−0.273 **	0.607 **	-					
8. Attachment Strength to Friend	−0.058	−0.162 *	0.060	0.054	−0.263 **	0.240 **	0.108 *	-				
9. Hierarchy vs. No Hierarchy	0.036	0.005	0.029	0.041	−0.139 **	0.140 **	0.239 **	0.114 **	-			
10. Physical Centrality	0.138 **	0.028	−0.058	−0.104 **	0.212 **	0.284 **	0.230 **	0.146 **	0.058	-		
11. Physical Density	0.114 **	0.021	−0.060	−0.094 **	0.152 **	0.258 **	0.269 **	0.093 *	0.063	0.736 **	-	
12. Attachment Strength Centrality	0.001	−0.058	−0.091 **	−0.080 *	0.563 **	0.675 **	0.645 **	0.586 **	0.171 *	0.290 **	0.288 **	-

*Note: n* = 657 (participants who rated romantic partners for attachment strength), *n* = 708 (participants who rated mothers for attachment strength), *n* = 545 (participants who rated fathers for attachment strength), *n* = 709 (participants who rated friends for attachment strength). * *p* < 0.05, ** *p* < 0.01.

**Table 3 behavsci-14-00647-t003:** Composition of collapsed category results for attachment figures based on age groups.

Age Group	*n*	Attachment Figure	*n*	% (Number of Times Selected as the Common Primary Target)
Early Adulthood (ages 22–34)	288	Romantic Partner	158	54.9
		Mother	83	28.8
		Father	36	12.5
		Sibling	33	11.5
		Friend	54	18.8
		Child	4	1.4
		Other	30	10.4
		Total	398	138.2 *
Early Middle Age (ages 35–44)	304	Romantic Partner	180	59.2
		Mother	82	27.0
		Father	42	13.8
		Sibling	33	10.9
		Friend	44	14.5
		Child	17	5.6
		Other	38	12.5
		Total	436	143.4 *
Late Middle Age (ages 45–64)	276	Romantic Partner	144	52.2
		Mother	58	21.0
		Father	20	7.2
		Sibling	38	13.8
		Friend	62	22.5
		Child	29	10.5
		Other	36	13.0
		Total	387	140.2 *
Late Adulthood (ages 65 and older)	62	Romantic Partner	35	56.5
		Mother	7	11.3
		Father	4	6.5
		Sibling	8	12.9
		Friend	14	22.6
		Child	11	17.7
		Other	32	51.6
		Total	111	179.0 *
Total	930			

*Note.* * Total percent exceeds 100 due to the nomination of multiple primary figures.

**Table 4 behavsci-14-00647-t004:** Sequential regression analyses of demographic variables on attachment strength to different attachment figures.

		Attachment Figure											
		Romantic Partner		Mother		Father		Sibling		Friend		Child	
Model	Predictor	*R*^2^ch	β	*R*^2^ch	β	*R*^2^ch	β	*R*^2^ch	β	*R*^2^ch	β	*R*^2^ch	β
1		0.009 *		0.034 **		0.019 *		0.004		0.029 **		0.077 **	
	Age		0.028		0.094 *		0.058		0.021		−0.054		0.195 *
	Parental Status		−0.097 *		−0.027		−0.081		−0.054		0.011		
	Romantic Status				0.167 **		0.135 **		0.053		0.157 **		0.240 **
2		0.001		0.004		0.001		0.005		0.001		0.015	
	Age x Parental Status		−0.044		−0.069		−0.025		−0.064		−0.013		
	Age x Romantic Status				−0.021		0.020		0.033		0.037		−0.196
	Parental Status x Romantic Status				0.016		0.047		0.075		0.041		
*R* ^2^		0.010		0.038		0.021		0.008		0.031		0.092	
Adjusted *R^2^*		0.006		0.030		0.010		−0.002		0.023		0.072	
*F_(df)_*		2.276 _(3, 657)_		4.583 ** _(6, 708)_		1.904 _(6, 545)_		0.838 _(6, 594)_		3.723 ** _(6, 709)_		4.519 ** _(3, 138)_	

*Note.* Parental status dummy coded as 0 = no child, 1 = child; romantic status dummy coded as 0 = no romantic partner, 1 = romantic partner. * *p* < 0.05. ** *p* < 0.01. R^2^ch = R^2^ change.

**Table 5 behavsci-14-00647-t005:** Descriptive statistics for physical proximity to different figures.

Attachment Figure	*n*	M *(miles)*	SD	0–10 Miles (%)	11–250 Miles (%)	251–500 Miles (%)	>500 Miles (%)
Romantic Partner	657	6.11	60.45	520 (79.1)	75 (11.4)	37 (5.6)	25 (3.8)
Mother	708	182.71	70.49	296 (41.8)	222 (31.4)	126 (17.8)	64 (9.0)
Father	545	216.58	69.66	163 (29.9)	191 (35.0)	110 (20.2)	81 (14.9)
Sibling	594	197.15	61.65	192 (32.3)	232 (39.1)	108 (18.2)	62 (10.4)
Friend	709	188.29	66.78	255 (36.0)	267 (37.7)	131 (18.5)	56 (7.9)
Child	138	8.19	63.86	102 (73.9)	21 (15.2)	10 (7.3)	5 (3.6)

**Table 6 behavsci-14-00647-t006:** Sequential regression analyses of demographic variables on physical proximity to different attachment figures.

		Attachment Figure											
		Romantic Partner		Mother		Father		Sibling		Friend		Child	
Model	Predictor	*R*^2^ch	β	*R*^2^ch	β	*R*^2^ch	β	*R*^2^ch	β	*R*^2^ch	β	*R*^2^ch	β
1		0.026 **		0.069 **		0.035 **		0.034 **		0.028 **		0.248 **	
	Age		0.118 *		0.199 **		0.079		0.069		0.039		0.560 **
	PS		−0.130 **		0.047		−0.077		−0.114		0.039		
	RS				0.248 **		0.169 **		0.182 **		0.209 **		−0.029
2		0.000		0.006		0.005		0.000		0.006		0.005	
	Age x PS		0.006		−0.021		−0.079		−0.011		−0.056		
	Age x RS				0.051		0.069		0.010		0.012		0.114
	PS x RS				−0.118		0.006		−0.003		−0.107		
*R* ^2^		0.026		0.075		0.041		0.035		0.033		0.253	
Adjusted *R^2^*		0.021		0.067		0.030		0.025		0.025		0.236	
*F_(df)_*		5.722 ** _(3, 657)_		9.486 ** _(6, 708)_		3.790 ** _(6, 545)_		3.502 ** _(6, 594)_		4.021 ** _(6, 709)_		15.206 _(3, 138)_	

*Note.* PS = parental status; RS = romantic status; parental status coded as 0 = no child, 1 = child; romantic status coded as 0 = no romantic partner, 1 = romantic partner. * *p* < 0.05. ** *p* < 0.01. R^2^ch = R^2^ change.

**Table 7 behavsci-14-00647-t007:** Sequential Regression Analyses of Demographic Variables on the Centrality and Density of Attachment Networks.

		Morphology Factors							
		Degree of Network Hierarchy		Emotional Centrality		Physical Centrality		Physical Density	
Model	Predictor	*R*^2^ch	β	*R*^2^ch	β	*R*^2^ch	β	*R*^2^ch	β
1		0.049 **		0.007		0.034 **		0.025 **	
	Age		0.029		0.027		0.164 **		0.092 *
	Parental Status		−0.135 **		−0.125 *		−0.073		−0.089
	Romantic Status		0.163 **		−0.057		0.030		0.016
2		0.007		0.003		0.006		0.002	
	Age x Parental Status		−0.062		0.035		−0.061		−0.049
	Age x Romantic Status		0.026		−0.003		0.072		−0.010
	Parental Status x Romantic Status		0.152 *		0.085		−0.081		−0.038
*R* ^2^		0.056		0.010		0.040		0.027	
Adjusted *R^2^*		0.050		0.003		0.034		0.021	
*F_(df)_*		9.185 ** _(6, 930)_		1.529 _(6, 930)_		6.472 ** _(6, 930)_		4.268 ** _(6, 930)_	

*Note.* Degree of network hierarchy coded 0 = no hierarchy, 1 = hierarchy; parental status coded 0 = no child, 1 = child; romantic status coded 0 = no romantic partner, 1 = romantic partner.* *p* < 0.05. ** *p* < 0.01. R^2^ch = R^2^ change.

## Data Availability

The data that support the findings of this study are available from the corresponding author, [HF], upon reasonable request.

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
