# Peer review of "“All You Need Is Love” a Social Network Approach to Understanding Attachment Networks in Adulthood"

_behavsci, 2024, doi:10.3390/bs14080647_

Round 1

Reviewer 1 Report

Comments and Suggestions for Authors

This was a really interesting article to read – the study provides important nuanced insight to attachment networks and the development of a new measure that captures the complexity of adult attachment and attachment figures. It accounts for adults who are in romantic relationships and the impact of parenthood on attachment, as well as age and older adults. Discussion of existing measures has been executed well, leading coherently to the need for the WHMT, which was explained very clearly and well justified in terms of its ability to measure the five dimensions of adult attachment and the three dimensions of attachment network morphology. The article has been very well written, and was a pleasure to read. Thank you for providing me with the opportunity to review this paper.

The manuscript has been written clearly, with concepts explained well and structured in a coherent manner. It is an extremely relevant study for the field of adult attachment and attachment measures to better capture its complexity and fluidity. Citations are appropriate and the results seem reproducible as a result of a detailed Method section. The tables are clear and easy to follow/understand, and results have been discussed well in the context of existing theoretical positions regarding attachment figures, and previous research. The study is ethically sound.

A few minor points given in the spirit of further enhancing readability and understanding for the readership:

Page 5, lines 223-224: emerging adults (24-34 years old) are mentioned, however Table 1 points to Early Adulthood (ages 22-34) with a discrepancy in ages between these. Furthermore, the literature on Emerging Adulthood (EA) tends to divide the ages for EAs into either 18-25 years old (e.g. see Arnett), or 18-30 years old, with 30-40 years referred to as ‘young adulthood’ in some literature. I recommend the authors define what they mean by emerging adulthood here, or omit stating this, to help with clarity and help avoid potential confusion in these constructs.

Page 5, line 227: the authors mention ‘Amazon’s Mechanical Turk (MTurk workers), but not all readers will be familiar with what this is. I recommend the authors include a brief explanation of what this is, and make a bit clearer why this was the selected method of recruitment.

Same re ‘HIT’ on page 5, line 236.

Page 6, lines 253-254: I wonder whether the authors would like to place quotation marks around the participant instruction, e.g. “how close you want to be to that person when you are feeling emotionally insecure, unprotected or vulnerable” so that this is clear, especially as the person changes to the second person.

Page 6: As the reader, I would really value having a visual in the form of a figure of the concentric circles, or a screenshot example of these with the icons on the diagram either of the networks structure or of the physical distance, or ideally both, e.g. the second one following on from the first one. This could add visual impact and understanding of the value of the WHMT.

Page 10, lines 417-418: As this sentence refers to the findings being consistent with previous research, citations of this previous research should be included here.

Author Response

Thank you for your constructive feedback and suggestions to strengthen our manuscript.  The revised manuscript includes a new section on adult attachment theory, new figures to illustrate the WHMT diagrammatic measure, and revisions to improve the flow and clarity of concepts.  We also added new references to support of our findings and to connect our work to the current body of literature. Below we detail these changes with respect to each reviewer's comment. 

Reviewer 1. First, thank you for your encouragement and positive comments regarding the clarity and importance of our work.  We found your suggestions to significantly improve the clarity and flow of our manuscript.   

Suggestions for revision.

  1. Identifying “emerging adults” as a developmental period in our study.

Thank you for this observation.  We agree with the reviewer that our age boundaries do not fit the accepted time period of emerging adulthood.  We now use early adulthood in its place on page 5, lines 268-269, and no longer employ emerging adulthood at any point in referencing our sample. 

  1. Include a brief explanation of Amazon Mechanical Turk and of a “HIT”

Thank you for this suggestion. We now incorporate a brief explanation of both MTurk and a HIT:  See page 5, lines 273-276, and page 6, lines 287-290. 

  1. Place quotation marks around the participant instruction,”how close you want …”.

Resolved – see page 6, lines 308-309

  1. I would really value having a visual in the form of a figure of the concentric circles, or a screenshot example of these with the icons on the diagram either of the networks structure or of the physical distance, or ideally both.

Thank you for these suggestions.  We also feel the use of including a couple of figures to illustrate the WHMT diagrams is useful for both the “vulnerable self” diagram and the “physical distance diagram”.  We have included these two figures.  See pg. 7 & pg. 8.

  1. lines 417-418: As this sentence refers to the findings being consistent with previous research, citations of this previous research should be included here.

Thank you for identifying this oversight.  Resolved.  See page 12, line 481.

Reviewer 2 Report

Comments and Suggestions for Authors

First, thank you for the opportunity to review this work.

The manuscript makes a valuable contribution to understanding the structure of the attachment network in adults by using the WHMT to measure five dimensions: composition, physical distance and three dimensions of network morphology, including hierarchy, centrality and density. The work has three main objectives: to replicate existing findings on the composition of the attachment network; the romantic partner is the primary attachment figure in adulthood; and parental and romantic status is related to the strength of attachment to friends and parents.

The methodology is sound, and the results are presented clearly. This manuscript meets the journal's criteria. I will detail the main criticisms in the following.

The introduction is well-highlighted and cites relevant existing literature, providing a theoretical basis for the study. However, it could benefit from a more explicit articulation of the concept of attachment from a historical and research perspective by better delving into the concept of adult attachment.

E.g.

Hazan, C., & Shaver, P. (1987). Romantic love conceptualized as an attachment process. Journal of Personality and Social Psychology, 52(3), 511–524. https://doi.org/10.1037/0022-3514.52.3.511

La Guardia, J. G., Ryan, R. M., Couchman, C. E., & Deci, E. L. (2000). Within-person variation in security of attachment: A self-determination theory perspective on attachment, need fulfillment, and well-being. Journal of Personality and Social Psychology, 79(3), 367–384. https://doi.org/10.1037/0022-3514.79.3.367

Perazzini, M., Bontempo, D., Giancola, M., D'Amico, S., & Perilli, E. (2023). Adult Attachment and Fear of Missing Out: Does the Mindful Attitude Matter?. Healthcare (Basel, Switzerland)11(23), 3093. https://doi.org/10.3390/healthcare11233093

It would also be good to improve the statement of the problem this study intends to address. A clear definition of this gap would strengthen the justification for the current research.

The discussion section effectively summarises the findings, drawing 

The discussion should be explicitly more connected to the existing literature, strengthening the contribution of the current study to the broader field. In addition, a more thorough and supported exploration of the findings enhances the implications the study seeks to define: what is the aim?

I invite the authors to implement the limitations discussed in the text. 

For example, the choice of rating scales could be better discussed within the limitations.

Author Response

Thank you for your constructive feedback and suggestions to strengthen our manuscript.  The revised manuscript includes a new section on adult attachment theory, new figures to illustrate the WHMT diagrammatic measure, and revisions to improve the flow and clarity of concepts.  We also added new references to support our findings and to connect our work to the current body of literature. Below we detail these changes with respect to your comments. 

Reviewer 2. 

           Suggestions for revision.  

  1. Address the History and theory on adult attachment

Thank you for this suggestion.  We agree that some additional theory on the study of adult attachment helps strengthen this section.  We added a new section on the history and conceptualization of adult attachment incorporating Hazan-Shaver study, as well as four additional foundational papers.  See page 2, lines 60 to 79.  We did not include two additional articles that were suggested only because these papers focus more on using adult attachment quality (secure-insecure) to predict behavioral outcomes and not on the formation of adult attachments or attachment networks.  We found the FOMO study to be most interesting with clear implications for how adult attachment networks influence mental health and adjustment, but this outcome is still afield from the aim of the current study. 

  1. Improve the statement of the problem this study intends to address.

Thank you for this suggestion.  We added a statement to clarify how the aim of our study addresses a problem in attachment network research.  See page 5, lines 235-239.  In addition, we specify the problem and aims earlier on 1, lines 38 to 43.     

  1. The discussion should be explicitly more connected to the existing literature, strengthening the contribution of the current study to the broader field.

Thank you for this suggestion.  We revised and reorganized sections of the discussion, especially in relation to the replication findings, by making connections to classic theory and contemporary evidence on network composition.  See page15, lines 582-584 and lines 589-593, & 604. See also pg. 16, lines 644- 653.      

  1. The choice of rating scales could be better discussed within the limitations.

Thank you for this suggestion, however, we are asking for some clarification on this point.  Do you mean that we should discuss the rating scales we used in our limitation section, and if so – what rating scales specifically?  Alternatively, do you mean that we should expand our discussion on the limitation of other studies that have used rating scales to assess adult attachment networks?

Round 2

Reviewer 2 Report

Comments and Suggestions for Authors

Dear authors, 

Thank you for responding to my comments and suggestions.  

Upon reading it, I found the requested changes more detailed and described. 

Best regards